# Responding to Climate Change in Small and Intermediate Cities: Comparative Policy Perspectives from India and South Africa

**David Simon** [1,*] , **Yutika Vora** [2] , **Tarun Sharma** [2] **and Warren Smit** [3]

[1] Development Geography, Department of Geography, Royal Holloway, University of London, Egham, Surrey TW20 0EX, UK

[2] Nagrika, Rajpur Road, Dehradun 248001, India; yutika@nagrika.org (Y.V.); tarun@nagrika.org (T.S.)

[3] African Centre for Cities, University of Cape Town, Rondebosch 7701, South Africa; warren.smit@uct.ac.za

* Correspondence: d.simon@rhul.ac.uk

**Abstract:** Remarkably little is known about how small and intermediate urban centres tackle their various sustainability challenges, particularly climate and broader environmental change. Accordingly, we address this in the very different contexts of India and South Africa. We conceptualise the small and intermediate towns, and the policy challenges and priorities for mitigating and adapting to the effects of climate/environmental change that can enable transformative adaptations to changing conditions. Central issues are the divisions of powers, responsibilities and the fiscal capacity and independence of local authorities within the respective countries' multi-level policy and governance frameworks. In India, various functions have been constitutionally devolved to city governments to enable them to govern themselves, while more strategic ones lie at state level. In South Africa, the divisions of power and responsibility vary by city size category. We compare the relevant city government functions in each country and how they can enable/disable policy responses to climate change. The relationship between their sustainable development strategies, plans, budgets, and actions are assessed and illustrated with particular reference to Thiruvananthapuram, Shimla and Bhubaneswar in India and Drakenstein, George and Stellenbosch in South Africa.

**Keywords:** small and intermediate towns; South Africa; India; climate change; urban sustainability; multi-level governance; Thiruvananthapuram; Shimla; Bhubaneswar; Stellenbosch

## 1. Introduction

As a contribution to this Special Issue's focus on sustainable urbanization in rural regions, this paper addresses the surprisingly persistent gap in our knowledge about how the many sustainability challenges facing small and intermediate urban areas are understood and managed. In particular, it examines governance and policy responses to climate change and broader environmental challenges in small and intermediate towns and cities in the very different contexts of India and South Africa.

The complexity of urban dynamics, and diversity of conditions across the numerous small and intermediate cities in each country, must be recognized at the outset. Moreover, it is often difficult to isolate climate and broader environmental changes from the broader matrix of ongoing changes and challenges facing small towns [1], but local contexts are always important considerations [2] (pp. 132–133). Inevitably, therefore, it is impossible to be entirely comprehensive and inclusive. Hence, we address the broad situation and general trends affecting small and intermediate cities in each country, before illustrating the issues in greater detail by means of selected case studies.

Within the context of the known range and likely parameters of climate/environmental change in different biophysical and anthropogenic settings, we examine how national, state/provincial, and local narratives of climate change are articulated in each country.

Furthermore, we conceptualise the challenges and priorities for mitigating its effects and undertaking transformative adaptations to changing conditions and building resilience in the diverse, resource-constrained contexts of these two countries.

How these challenges are addressed and resources are provided—and the extent to which funds match priorities—reflects the divisions of powers, responsibilities and the fiscal independence of local authorities within the respective countries' multi-level governance frameworks. In India, various functions have been constitutionally devolved to urban governments to enable them to govern themselves, while more strategic ones lie at state level. In South Africa, the divisions of power and responsibility vary by city size category. We will compare the relevant city government functions in each country and how they can enable/disable various communities and individuals in their response to climate change. Key issues include the low human and financial capacity of many local governments, the diversity of climate change plans and action in relation to city size, and the diverse potential to map and have a positive impact on their carbon footprints through coherent action. The relationship between their sustainable development strategies, plans, budgets, and actions will be assessed and illustrated with appropriate examples.

We have chosen these countries because they are two of the BRICS grouping of dynamic regional economies, comprising Brazil, Russia, India, China, and South Africa, which are seeking to act jointly to reshape global trade and mutually beneficial relationships [3,4]. Although the coherence of this group has declined in the wake of the financial crisis and COVID-19 pandemic [5], this cross-continental comparative assessment understands this as part of the broader context to urban trends. This comparison has added interest because, as regional powers with long traditions of urban governance and considerable but highly unevenly distributed institutional capacity and resourcing, India and South Africa are often regarded as countries able to address climate/environmental change autonomously. As such, they are likely to be "early adopters" of innovative approaches that neighbouring countries could seek to emulate and adapt.

In fact, the national policy frameworks on climate change have evolved comparably in both countries over the last 10–15 years. While India released its National Action Plan on Climate Change (NAPCC) in 2008 [6,7], South Africa published its National Climate Change Response Strategy (NCCRS) in 2004, followed by a Long-Term Mitigation Scenario to mitigate its emissions in 2008 [8]. Following up on the NCCRS, South Africa approved its National Climate Change Response White Paper (NCCRWP) in 2011, which eventually provided the key framework to South Africa's national climate policy. In 2012, climate change became part of the National Development Plan, the holistic development plan designed for the country [9]. Similarly, in 2012, India followed up the NAPCC with financial allocations to support implementation of the various components of NAPCC in its 12th Five Year Plan, equivalent of South Africa's National Development Plan [10].

## 2. Materials and Methods

This paper is based on policy analysis, using case studies from India and South Africa. Policy analysis is a well-established method in the social sciences [11–13] as "a process of multi-disciplinary inquiry aiming at the creation, critical assessment, and communication of policy-relevant information" [11] (p. 2). It has been recognized "as a method for structuring information and providing opportunities for the development of alternative choices for the policymaker" [14] (p. 6). Policy analysis is an effective tool to inform policy formulation by identifying the "capabilities and resources" of the policy makers to formulate and implement policies. The present article analyses the capacity of small and intermediate cities in India and South Africa, to implement policies that can respond to the challenges of climate change [15].

Policy documents and regulations formulated to give them effect are essential research materials in relation to governance and policy analysis, because they reflect how scientific (including social scientific) evidence and knowledge are understood by officials and mediated by political considerations to frame official responses. In complex, multi-faceted,

and rapidly changing fields, such as climate/environmental change and sustainability, key issues include how well the issues are understood and "translated" into policy and guidelines for implementation, how effectively this occurs across the various sectors or divisions within governance institutions (in this case urban local authorities) to ensure coherence and consistency, and how adequate the resourcing is.

As explained below, small and intermediate urban areas rely on state/provincial/regional and also national governments for a high proportion of financial and human resource capacity, while powers and responsibilities are shared in diverse ways—known as multi-level (or multi-scale) governance. Indeed, national governments also increasingly align their policies with global governance agreements and conventions, not least on climate/environmental change. Hence, it is essential to examine the documentation at all levels. This also facilitates the qualitative cross-country comparison, which we develop in Section 4.

The aim of case study research is to use a mix of methods to analyse processes within their context [16]. Thus, case studies are an important part of policy analysis for analysing issues that may be of general public interest and/or represent "nationally important" issues relevant to policy process or practitioners [17]. Yin [18] outlines the various reasons for a choice of a case study. For the Indian case study, we utilised secondary sources relevant to the policy and governance ecosystem in India, focused particularly on the environmental governance aspects of cities. The perspectives are also informed by extensive empirical research work by two co-authors in Indian states on the capacity of city governments relating to various municipal functions, functionaries, and financing. We also undertook a review of some of the key policies governing Indian cities, including the constitutional provisions, relevant state level municipal acts, and key policy guidelines of select cities. This includes a policy review of a constitutional amendment pertaining to municipal functions; three state municipal acts; three state level climate change action plans; multiple national policies on clean air including national level climate change action plan and its eight submissions; a national clean air program; relevant provisions of the 12th Five Year Action Plan; and various policy developments reviewed in journal articles and news reports.

The South African section is based on a review of relevant policy documentation and academic literature relating to small cities, local government, and sustainability in South Africa. Stellenbosch was chosen as an "extreme" or "unique" case [18], in that it is has the highest levels of poverty and inequality of the three small cities in the Western Cape and is also arguably in the most fragile environmental setting. Key policy documents and academic papers relating to sustainability issues in Stellenbosch (and small cities and issues of sustainability in South Africa more broadly) were identified and analysed. This included a total of 12 relevant academic publications from the 2010–2020 period, three national policy documents, and the four key Stellenbosch Municipality policy documents of relevance to sustainability issues: the Integrated Development Plan, Spatial Development Framework, Environmental Management Framework, and the Drought Response Plan.

### 2.1. Definitional Issues

Small towns are commonly regarded as having populations of between 20,000 and 100,000 [1]. The United Nations defines small and intermediate cities as having a population of under 500,000 inhabitants. While this is widely appropriate, including for South Africa, the scope and scale of urbanization in China and India, both of which now have numerous megacities (over 10 million inhabitants), multi-million and other large cities, is such that this ceiling would exclude many urban areas that function as intermediate cities within their national urban systems. Accordingly, to accommodate the Indian context, we adopt a pragmatic working definition of small and intermediate cities having a population of up to 750,000 inhabitants.

The respective national census definitions of what constitutes an urban area emphasise the point. In India, the minimum threshold for an urban area is a population of at least 5000, living at a density of at least 400 per km$^2$, and with at least 75% of the male adult

population engaged in non-agricultural work. This is a complex and demanding multi-faceted definition, of which the population threshold is just one element. Hence, numerous large villages would not qualify if agriculture were the main or a large livelihood sector. Although places administered by a municipal corporation or other such body are also automatically defined as urban, this also helps explain why India's population was only 31.2% urban at the 2011 census [19]. Such a definition would be unworkable in most other countries. Within this narrow definition though, almost 60% of India's urban population lives in cities with a population of less than one million.

Since the 2001 census, South Africa has avoided a specific urban definition in favour of a morphological and function definition based on the classification of dominant enumeration area types [20]. The population was 62.75% urban in 2011, rising to an estimated 66.86% in 2019 [21]. The eight largest cities are defined by the government as metropolitan areas (and are governed by "metropolitan municipalities"), and the South African government's Cities Support Programme classifies the next rank of 22 smaller cities (defined by a combination of population size, economy size and municipal budget size) as "secondary cities" [22]. Subsequent work on small cities in South Africa has used the same definition [23,24]. These cities have populations of between about 100,000 and 700,000.

### 2.2. Conceptual Approaches

**Small Cities and Policy Autonomy:** In so far as large cities tend to have greater economic diversity and specialization, as well as revenue from property and other local taxes, urban size often serves as a proxy for assumed urban functional specialization and governance capacity. Despite the diversity of conditions across the range of small and intermediate cities within and between countries, they do generally lack substantial autonomous capacity to formulate policies and administer them. Hence, they tend to fall under the jurisdiction of larger institutions or sub-national authorities that provide oversight, key skills, and resources. Precise arrangements differ because of the divisions of powers, responsibilities, and resources between national, regional, and local governments in each country; the contrasts between India and South Africa on this score, for instance, illustrate the point clearly.

**Small Cities within National Policy Frameworks:** Conceptually, the position and role of small towns and intermediate cities within national urban and urbanisation policies has changed over time according to prevailing academic ideas, political ideologies, and policy conventions. During the heyday of modernisation theory, for instance, from the late 1950s to mid-1970s, the emphasis was on innovation diffusion from large urban cores to the periphery, and often inappropriate and wasteful attempts to promote rural development through large-scale investment. With the advent of basic needs and the more participatory approaches from the 1970s, the focus shifted to smaller-scale and appropriate investments, linked to such urban centres as key nodes for integrated rural development. Under more authoritarian and state socialist regimes, rhetoric of rural development through mechanisation often sat awkwardly with the realities of top-down political control. At the lower end of the urban spectrum, small towns in rural areas have often served as farmer and related service centres, with little economic dynamism or development potential. In more urbanised regions, they have generally served little role except as trading posts and wholesale aggregation points for city-bound produce and distribution nodes for manufactured goods and services. Such national policies were inevitably diverse—at least 11 types could be distinguished [25] (pp. 238–246).

**Small Cities and Recent Policy Shifts:** Since the end of the Cold War and the rolling back of the state under supposedly neo-liberal policies, increased ideological pluralism and market orientation has driven major policy shifts. One consequence was that often-discredited and ineffective national urban policies were widely—but not universally—discarded in favour of sub-national and local urban entrepreneurialism—strategies which proved effective for many large cities with national and global ambitions but were unachievable for small and less distinctive towns and cities [25–27].

More recently, national urban policies have experienced a revival in view of the need for co-ordination of investments and the challenges of increasingly complex national and international agendas for achieving sustainable development (see Table 1), including the 2030 Agenda and its Sustainable Development Goals (SDGs), Sendai Framework for Disaster Risk Reduction, and New Urban Agenda, all adopted in 2015–16. Indeed, the number of (very diverse) countries formulating or already implementing new urban policies exceeds 100 and is growing apace, encouraged by the United Nations and OECD [25,28,29].

The extent to which, and how, climate change and related challenges are now being addressed in small and intermediate cities varies considerably but reflects the above issues. Of particular importance are evolving national urban and climate/environmental change policies; the distribution of powers, responsibilities and resources within multi-level governance arrangements in each country [30]; the challenges of transboundary collaboration among adjacent local authorities of different categories and capacities that include small or intermediate towns [31]; and the extent to which countries in receipt of official development assistance (ODA) from donor governments and intergovernmental organisations are able to tap dedicated funds to promote climate mitigation, adaptation, or transformative change. Indeed, UN-Habitat and the OECD, among others, have produced guidance for incorporating global sustainable development agendas and climate change into national spatial and urban policies [32,33].

Conceptually, small towns may have particular advantages in relation to often close and organic relationships with their immediate hinterlands, opportunities for marketing, and receipt of inputs if located on or near major transport routes, and the potential to generate political will at this more personal scale. Conversely, resources are often constrained, remoteness from transport networks often exacerbates trade and other flows, poverty, and inertia, while political and economic leaders may be sceptical about or opposed to explicit climate action, making mobilisation more difficult [1]. We turn in the following sections to detailed examination of these policies and processes in small and intermediate cities in India and South Africa.

## 3. Results

### 3.1. Perspectives from India

3.1.1. India's Urban Context

India had an urban population of 377 million as per the 2011 census, which made India's population 31% urban. In context of the definitions discussed in the above section, there are 4041 statutory towns, the cities which are governed by a municipal government created by a legal statute. There are also 7935 census towns, meaning that they have urban characteristics, as highlighted in the section above. Keeping the definitional issues aside, urbanisation is concentrated in relatively few cities. Almost 70% of the urban population lives in 468 Class I towns which have a population of more than 0.1 million [34,35].

**Table 1.** Key features of Thiruvananthpuram, Bhubaneswar and Shimla municipalities.

| Municipality | Population (2011) | Gini Coefficient (2016) | Proportion Living in Informal Housing (2011) |
|---|---|---|---|
| Thiruvananthpuram | 1,687,406 | 0.39 | 18.52% |
| Bhubaneswar | 881,988 | 0.35 | 0.42% |
| Shimla | 171,817 | NA | NA% |

Source: Based on refs. [36,37].

Furthermore, this urbanisation paradigm is marked by a relative decline of small cities in the urban system. There has been a "consistent strengthening" of large cities in India's urban system at the expense of decline in the share of urban population living in small and medium cities. This has led to unequal distribution of urban population among different city size categories. Almost 43% of the total urban population lives in the larger million-plus cities [35]. Nevertheless, almost 60% of India's urban population still lives in

cities with fewer than one million people—approximately the size used for the working definition of small and intermediate cities we are considering in this paper. We have chosen three representative small cities in different parts of India for our analysis here, namely Thiruvananthpuram (formerly Trivandrum), the state capital of Kerala, Bhubaneswar in Odisha state, and Shimla, the state capital of Himachal Pradesh (see Figure 1 and Table 1).

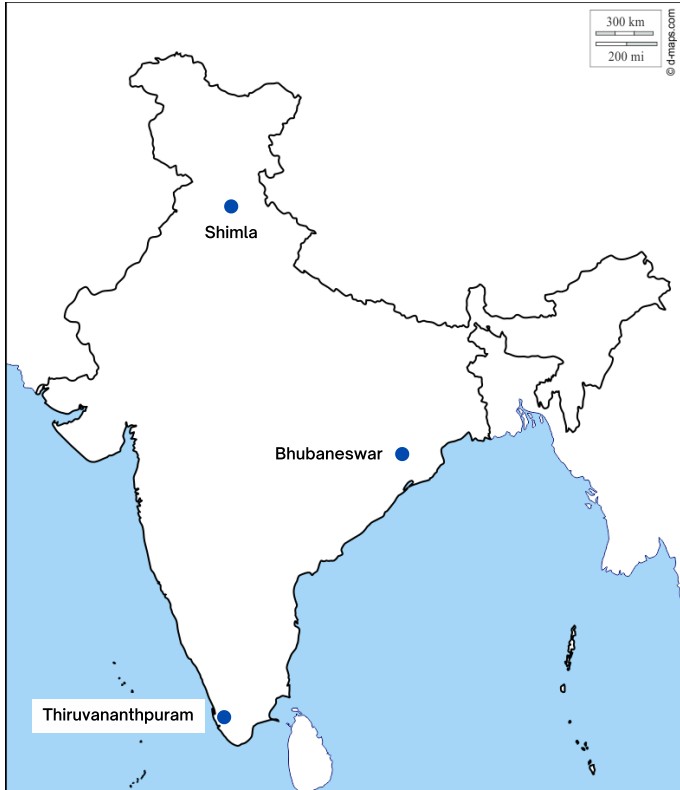

**Figure 1.** Location of the Indian cities.

### 3.1.2. Urban and Environmental Governance

In terms of India's constitution, governance of cities is a state responsibility, i.e., state governments enact laws and create policies to govern their cities and there is no national urban policy framework to guide the urbanisation process at national level. Union government programmes, however, provide the guidance and reflect the national priorities for urbanisation [38].

Governance was further devolved to city level governments/municipalities by a constitutional amendment in 1994. However, this devolution did not occur to the desired extent. The declining importance of smaller cities in the urban systems has also been accompanied by this lack of local power to formulate policies and implement programmes, especially relating to mitigating or adapting to the impact of climate change. The 74th Amendment of the Constitution in 1992 paved the way for decentralising 18 key functions to the municipalities. Twenty-five years on, many of these functions are still not performed by them, but instead by various provincial or national government agencies.

In most Indian cities, policies that govern urban or spatial planning and regulate land use fall under the jurisdiction of parastatal agencies such as the Town and Country Planning Organisation, Development Authorities or Urban Improvement Trusts [39]. Nagrika analysed the level of devolution of these functions, including those that impact the capability of municipalities to deal with climate change. Table 2 provides a summary of three small-sized cities with respect to their jurisdiction over key functions related to climate change.

Table 2. Jurisdictions of three small-sized Indian cities over key climate change-related functions.

| | Regulation of Land-Use | Water Supply | Sanitation and Solid Waste Management | Urban Forestry and Environmental Protection |
|---|---|---|---|---|
| Thiruvananthapuram | Town and Country Planning Department is responsible for land use zoning. Municipality can only approve building plans for smaller buildings | Performed by Kerala Water Authority. Some other maintenance functions are performed by municipality | Municipality is responsible for collection and disposal of solid waste and sanitation in the city | State level forest officers are responsible for this function |
| Shimla | Town and Country Planning Department is responsible for regulating land use | Performed by a company jointly promoted by municipal corporation and the state government. In other cities, this function is performed by the state-run Department of Irrigation and Public Health | Municipality is responsible for collection and disposal of solid waste and sanitation in the city | State's forest department. Municipal Act has provision for a tree officer to count and allow felling of trees, but the person is not appointed |
| Bhubaneswar | Performed by Bhubaneswar Development Authority | Performed by the Public Health and Engineering Organisation (PHEO), which is under the administrative control of state government | Municipality responsible for collection and disposal of solid waste and sanitation in the city | Municipality mostly responsible for parks and gardens. Plantations are co-ordinated with state forest department |

As is evident, except for the management of solid waste, many of the other functions which can equip smaller cities to deal with climate change are not fully within their jurisdiction. Urban planning and regulation of land use is one such important function. While the function itself lies with state-level agencies, the availability of skilled professionals, such as planners, is also limited. Indeed, there are only an estimated 0.23 planners per 100,000 people in India [40]. Along with constraints on the functions and capacity to perform these functions, funding is also limited for smaller local governments. The own-source revenues of these cities are inadequate to finance "large scale climate friendly investments" and they are largely dependent on external funding such as "grants, subsidies and international climate and development finance", which come with their own conditionalities and leave little room for local action [40].

There are no institutional mechanisms such as policy mandates or budgeting outlays at the city level that are aligned to climate change action, including national and international frameworks. While the cities do not budget any funds for policy measures related to climate change from their own resources, they are often part of national and international climate programmes [41]. For example, two of the three Indian cities considered here (Shimla and Thiruvananthpuram) were part of the "Govt of India-UNDP Climate Risk Management Project in Urban Areas through Disaster Preparedness and Mitigation", which was a UNDP project that received financial assistance from USAID. Similarly, much of the funding to support cities' actions on clean air comes from national level budgetary resources. At the beginning of 2020, the Union Budget made a commitment of INR 44 billion for "formulating and implementing plans for ensuring cleaner air" in million-plus cities and towards the end of the year; the Ministry of Finance released half of that amount to build the capacity of cities to fund air quality measures [42]. A publication by UN-Habitat on National Urban Policies also affirms that the weak capacity of city governments has led to "haphazard and unplanned development" and the planning process did not address the

issues relating to provision of infrastructure, environmental conservation and financing, especially infrastructure financing [38].

As highlighted in the introductory section, the national policy framework on climate change has been shaped by the NAPCC. Within the NAPCC, there are eight missions which have been shaped around India's development needs such as a national solar mission, water mission, and Mission on Himalayan ecosystem, among others. The policies and actions of the NAPCC and its various missions, however, are guided through State-level action plans with limited inputs from city governments [43]. Various other important functions, which are also critical to manage the impact of climate change, are not within the jurisdiction of municipalities. These include the management of air quality and providing environmental approvals. As per Nagrika's study, most of these functions lie with national and state level agencies. Air quality is monitored through the national- and state-level pollution control boards. The national- and state-level powers supersede those of the local governments in creating large scale projects with significant environmental impact, such as setting up airports, waste management plants, and metros systems, among others. Crucially, the local governments have no powers to assess or approve such environmental impacts.

The experience of India and its small cities demonstrates the constraints at the level of functions, functionaries, as well as funding, to deal with climate change. The immediate risks of climate change manifest themselves at the level of the cities. However, the ability to deal with such risks is also highest at the level of local governments if they are empowered adequately [44]. In the context of multi-level governance frameworks that deal with urban issues, it is imperative that the smaller cities are empowered to create and implement strategic plans to deal with climate change within such frameworks [45].

*3.2. Perspectives from South Africa*

3.2.1. The Western Cape's Urban Context

South Africa had an estimated population of about 60 million in 2020 [46], of whom approximately 67% lived in urban areas [23]. The Western Cape Province consists of the south-western part of South Africa. It has an area of 129,449 km$^2$ (approximately the same size as England) and a population of about seven million [46]. The Western Cape is the second most urbanized of South Africa's nine provinces, with about 90% of the population living in urban areas [23]. With about four million inhabitants, the City of Cape Town accounts for over half the province's population. The non-metropolitan part of the province is mainly agricultural and generally has a low population density. The three municipalities in the province classified as secondary cities by the South African government's Cities Support Programme are Drakenstein (Paarl), Stellenbosch and George (Figure 2). As shown in Table 3 below, they all have populations in the 100,000 to approximately 300,000 range and have high levels of income inequality as measured by Gini coefficients—Gini coefficients of more than 0.4 are considered to be "unacceptably high" [47] (p. 72).

**Table 3.** Key features of Drakenstein, George and Stellenbosch municipalities.

| Municipality | Population (2018) | Gini Coefficient (2017) | Proportion Living in Informal Housing (2016) |
|---|---|---|---|
| Drakenstein | 300,991 | 0.59 | 9.7% |
| George | 213,819 | 0.61 | 16.1% |
| Stellenbosch | 186,730 | 0.63 | 34.9% |

Based on refs. [48–50].

Stellenbosch has the highest level of inequality and largest proportion of residents living in informal housing of the three cities and reflects the sustainability challenges faced by small cities in the Western Cape. It was established by Dutch settlers in 1679 and is located in a fertile valley within a wine-growing region. The municipality is located within the Cape Winelands Biosphere Reserve, which was formally designated by the United Nations Educational, Scientific and Cultural Organization (UNESCO) in 2007 [51].

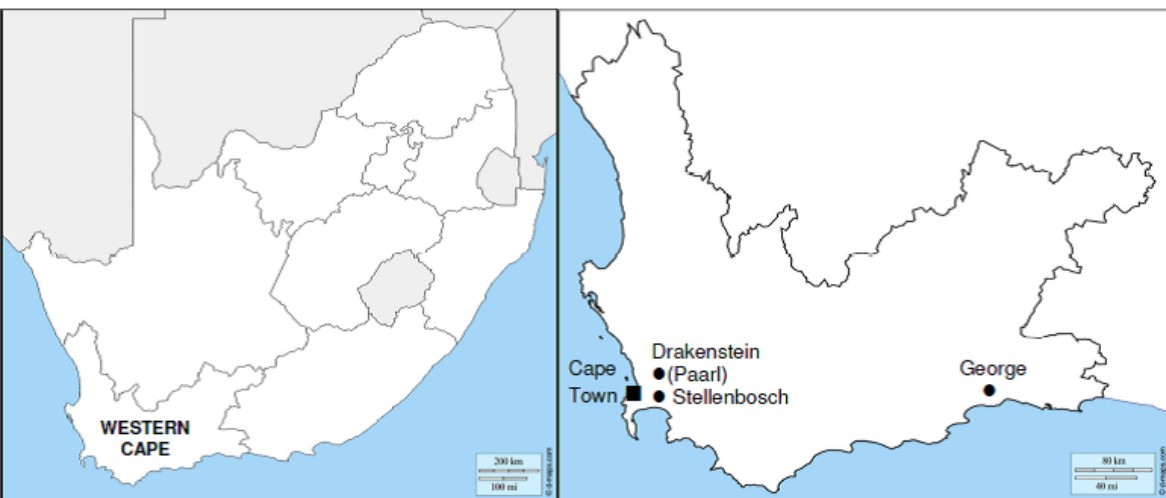

**Figure 2.** Location of the South African cities in the Western Cape.

Fairly rapid population growth in Stellenbosch has resulted in rapid spatial expansion and a housing crisis, manifesting in the growth of informal settlements. The population is growing at 2.2% per year, and currently approximately 35% of the population lives in informal housing without adequate services [52]. This rapid spatial growth has had a negative impact on the fragile natural environment of the area; Stellenbosch Municipality notes that "Development has meant the loss of many hectares of valuable agricultural land, and some pristine nature areas have been scarred . . . We have undermined a valuable biodiversity resource, not only as a context for tourism, but also as cultural heritage, a sacred space for healing, and the provider of valuable ecosystem services such as clean water, clean air, and erosion prevention" [52] (p. 113).

As with most other cities in the non-metropolitan part of the province, the local economy is largely based on agriculture and tourism, both of which are vulnerable to climate change. Water supply is also a major challenge because of low and uncertain rainfall. In 2018, Stellenbosch and neighbouring Cape Town faced a water crisis because of three years of drought [53], and water restrictions were introduced in Stellenbosch, limiting each person to a maximum of 50 litres per day [54]. As of 2018, about two thirds of Stellenbosch Municipality's water came from the Western Cape Water Supply System, a system of large dams and tunnels in the mountains to the east of Cape Town and Stellenbosch (one of which, the Berg River Dam, is within the Stellenbosch municipal area), operated by the National Department of Water and Sanitation, the Trans-Caledon Tunnel Authority, and the City of Cape Town [55]. The other third of Stellenbosch Municipality's water supply came from its own two Idas Valley dams; Stellenbosch Municipality subsequently also increased its supply of water from bore holes so as to decrease its reliance on external sources of water. Population growth and climate change will exacerbate this water challenge in the future. Electricity is another challenge. Eskom, the national electricity supplier, generates electricity through a national grid (mainly produced by coal-fired power stations in the north-east of the country), which sells the electricity to municipalities for sale to consumers. The use of coal is unsustainable in the long term, however, and Eskom has struggled to supply sufficient electricity, leading to frequent power shortages [56].

3.2.2. Urban and Environmental Governance

Non-metropolitan areas in South Africa have two tiers of local government, district municipalities and local municipalities. The responsibilities of local government are set out in the 1996 Constitution of South Africa [57]. These include the provision of most services (roads, water, sanitation, electricity, stormwater drainage, solid waste management, parks) as well as a range of other functions (planning, local economic development, tourism, environmental management, disaster risk management). Section 24 of the Constitution

states that everyone has the right to "have the environment protected, for the benefit of present and future generations, through reasonable legislative and other measures that: (i) prevent pollution and ecological degradation; (ii) promote conservation; and (iii) secure ecologically sustainable development and use of natural resources while promoting justifiable economic and social development" [57]. In addition to the Constitution, there are a range of other policies that attempt to address urban issues in South Africa, most notably the Integrated Urban Development Framework, but the implementation of these urban policies has been less than ideal [58].

In order to meet their many objectives, local governments in South Africa have a range of local sources of revenue, mainly property taxes and service charges (e.g., for water and electricity, the amounts charged for these services generally exceed the cost of providing them). There are also a range of conditional grants from the national and provincial government (mainly for housing and infrastructure), and local government also has access to a flexible share of national government revenue in terms of Section 227(1)(a) of the Constitution [57].

The Stellenbosch area falls under the Stellenbosch Municipality, which forms part of the Cape Winelands District. Hence, certain high-level functions are performed by the Cape Winelands District Municipality. Stellenbosch Municipality's Division of Spatial Planning, Heritage and Environment is primarily responsible for urban sustainability issues. The three most important policy documents of the Stellenbosch Municipality in terms of urban sustainability (as with other South African municipalities) are the five-year Integrated Development Plan, the Spatial Development Framework, and the Stellenbosch Environmental Management Framework.

The Stellenbosch Integrated Development Plan for 2017–2022 identifies five strategic focus areas for the Municipality, of which Strategic Focus Area 2 is a "Green and Sustainable Valley" [52]. In practice, however, the main focus is on other issues (such as the provision of services and local economic development), and very little of the budget is devoted to issues of urban sustainability; for example, only 2% of the 2017/2018 municipal budget was set aside for the "Green and Sustainable Valley" focus area [52]. The largest item of expenditure for this focus area is "Energy Efficiency and Demand Side Management", intended to reduce electricity use. A related municipal strategy is "Water Demand Management", focused on reducing water use. The Municipality notes that "In terms of adapting for climate change, water systems will need to be more robust and new or alternative sources of supply may need to be found" [52] (p. 77). The challenge with electricity and water demand management by South African municipalities is that the sale of these services are major sources of local government revenue, so effectively reducing demand for electricity and water would reduce local government revenue and probably require unpopular tariff increases (as became very evident in the neighbouring City of Cape Town during the water crisis of 2018). There are similar financial complexities for other elements of Stellenbosch's urban sustainability initiatives, for example, restrictions on spatial growth can prevent growth of income from property tax.

The Stellenbosch Spatial Development Framework includes an urban edge to discourage urban sprawl and identifies corridors for densification [59]. The Stellenbosch Environmental Management Framework sets out the environmental vision for the municipality as: "A municipality and communities that recognise the vital importance of their rich natural capital and manage these in a manner that ensures sustainability and fulfils the needs of all concerned" [60] (p. 14). The strategy particularly focuses on a set of guidelines to guide spatial development in an environmentally sustainable way. Broad awareness of the sustainability challenges facing Stellenbosch is currently very uneven [61], but there have been attempts to raise awareness about environmental sustainability issues in the area [62]. One particular pilot project in the Stellenbosch area, Lynedoch Eco-Village, has successfully demonstrated the practicality of a variety of innovative and sustainable technologies, including energy, water, sanitation and building materials [63].

The example of Stellenbosch shows that the municipalities of small cities in South Africa generally have the appropriate powers and strategies to address issues of urban sustainability. The challenges are with prioritisation and implementation, mainly caused by, firstly, the general lack of awareness of urban sustainability compared to challenges such as infrastructure provision and economic growth, and secondly, perverse financial incentives (such as income from the sale of electricity and water, and property taxes from new developments) that tend to perpetuate unsustainable urban patterns and practices. The fact that multiple levels of government are involved, and environmental issues cut across municipal boundaries, further complicates matters (this is particularly highlighted by the issues of water and electricity).

## 4. Discussion

Although often neglected in academic literature, small cities play an important role in the urban systems of both India and South Africa. The immediate risks of climate change mainly manifest themselves at the level of the cities. The various global and regional agreements that seek to manage the risks emanating out of climate change are increasingly acknowledging the role of city governments, including the Paris COP21, EU 2030 climate and energy policy framework, German climate protection plan 2050, as well as China's national climate change program [64]. Although India lacks a comprehensive national urban policy framework, the constitutional provisions envisioned a central role for municipalities in implementing programmes needed for environmental conservation, although cities have not been empowered to take up this role. The role of cities in national and state level policy frameworks, such as climate change action plans and clean air programs, is also limited. Similarly, in South Africa, the Constitution of 1996 envisioned a central role for municipalities in a range of developmental issues, particularly service delivery and urban planning and management. The recent South African national policy framework, the Integrated Urban Development Framework, translates the Sustainable Development Goals and a number of national policies into a set of guidelines to steer urban growth and management so as to achieve "Cities and towns that are well planned and efficient, and so capture the benefits of productivity and growth, invest in integrated social and economic development, and reduce pollution and carbon emissions, resulting in a sustainable quality of life for all citizens" [65] (p. 39).

Addressing climate change and environmental sustainability in smaller cities is, however, complex, because the smaller economies and small revenue bases of those cities often mean that local governments in small cities have considerably less capacity than is the case with local governments in big cities. Smaller cities are also sometimes crowded out by bigger cities in terms of resources, such as is the case in Stellenbosch, where much of the water collected in the municipal area is actually diverted to the neighbouring big city of Cape Town.

A major issue determining whether or not cities are able to address climate change and environment sustainability effectively is whether local government has appropriate powers and functions to address key issues that impact on sustainability. In South Africa, municipalities have the main responsibility for land-use regulation, water supply, sanitation, solid waste management and environmental protection within their municipal areas (although district municipalities and provincial government also plays oversight and support roles). In India, by contrast, municipalities are generally only responsible for sanitation and waste management and not for important functions such as land-use regulation, water supply, and environmental protection, which means that they are less well-equipped to deal with issues of climate change and environmental sustainability. Even though city governments are constitutionally designated with key environmental functions, they are not empowered with financial and human resources to perform them. Smaller city governments are even more disempowered and the "planning and governance" in such cities is bureaucratic and lacks accountability to citizens or elected representatives [66].

The South African example of Stellenbosch shows that the allocation of responsibilities, while important, is only part of what is needed for local government to be able to address issues of climate change and environmental sustainability. Firstly, current flows of funding often perpetuate unsustainable patterns of urban growth. For example, the supply of water and electricity is a major source of local government revenue; hence, reducing the demand for water and electricity to promote sustainability has severe financial repercussions for the solvency of municipalities. This implies a requirement for a re-examination of revenue sources and flows for local governments in relation to sustainability criteria. Similarly, property taxes and development charges from new developments are an important source of local government revenue, and municipalities are therefore incentivized to keep approving new developments. Property taxes, user charges, and charges from building approvals form the major proportion of own source revenue of Indian municipalities as well.

Secondly, the vast challenges of poverty and inequality facing cities in the Global South, and a general lack of awareness about climate change and sustainability, means that these issues are often not high priorities compared to, for example, job creation. In India, the constitutional function "Planning for Economic and Social Development", is largely performed by parastatals and city governments have very limited understanding of what the function entails. Municipalities are only responsible for implementing state- and national-level programmes aimed at economic development, such as urban livelihood programmes. Thirdly, as the examples from India and South Africa both show, multi-scalar (or multi-level) governance means that within each city there are multiple stakeholders involved, including the national government, state or provincial government, and a range of governmental agencies. These different organisations and agencies often have different perspectives and strategies, and co-ordination between them can be difficult to achieve. While national (and/or) state governments tend to create policies and programmes which aim to chase macro policy goals, local governments often respond to immediate local needs and function more as service providers than as strategic decision-makers.

## 5. Conclusions

This paper has provided a comparative analysis of the generic governance and policy challenges facing small and intermediate towns and cities in tackling climate and broader environmental change [1,44] within the respective national contexts of South Africa and India, two of the five BRICS group of emerging powers. The most profound difference relates to demography, namely India's vastly greater population size and scale of urbanisation. Hence, small and intermediate Indian cities range up to 750,000 or more inhabitants, whereas in most other countries apart from China, 100,000 provides an appropriate ceiling for small towns and 500,000 for intermediate cities. Ironically, however, South Africa's complex definition means that a few intermediate cities have populations above 500,000, in one case approaching 750,000, and thus providing a similarity with India.

There are significant differences in terms of the respective national urban policies [24,25,35,38,58] and the precise divisions of roles, responsibilities, and resources among the national, state or provincial, and local levels of government in each country. An important similarity is that control of these levels by different political parties can increase the challenges of undertaking collaborative multi-level or multi-scalar governance, the effectiveness of which is a prerequisite for tackling climate change and other transboundary phenomena [30]. However, even control by the same party may not ensure smooth collaboration if structural ambiguity or misalignment, resource constraints, and rivalries among the respective political leaderships intrude.

The same applies to horizontal transboundary collaboration among local governments within the same metropolitan, city region or functional region. After all, each such entity has fixed boundaries, and collaborative leadership by the state/provincial authorities may be required in order to ensure that they are all able to work together strategically for mutual benefit [31]. This may be easier within the Indian context because the states

exercise stronger relevant powers, responsibilities and resources than do South African provinces for all but the smallest urban centres.

Broadly speaking, within the constraints of state or provincial control—which may vary in precise nature across states or provinces—the resources and capacity of small and intermediate urban councils tend to increase somewhat with population and the proportion of total revenue raised locally. Inevitably, though, resource constraints (including of skilled personnel) and reliance on state or provincial skilled professional and financial resource capacity—which itself constitutes a particular kind of externality—are often severe. This lack of autonomous capacity emphasises the importance of prioritisation among competing demands and also inhibits the making of binding decisions with medium to long-term implications as required to promote transitions or transformations towards sustainability, including tackling extreme events and climate change.

That said, the role of dynamic leadership should not be overlooked—an inspirational and energetic mayor, commissioner or senior climate change champion can make a really positive contribution. Conversely, frequent leadership changes inhibit progress and can trigger immobility while new incumbents find their feet or hesitate to take bold initiatives, however important, in the face of climate change and other unconventional challenges, that might prove controversial. This is an issue in both countries analysed here. In the Indian system, municipal commissioners and their deputies are state appointees, while South African municipal mayors and councillors may change frequently through elections or political party redeployment of representatives elected on proportional representation tickets.

Our analysis identifies the need for more concerted and strategic approaches to equipping governments in small and intermediate towns and cities to deal with climate change. This includes enhancing personnel and financial capacity, which are key current constraints. More broadly, addressing urban inequality by promoting urban justice and resource redistribution—an essential component of sustainability and climate change strategies—falls into the category of policies vulnerable to changing leaderships and policies [2] (pp. 139–140). Despite (or maybe because of) the smaller scale and more immediately visible nature of climate change impacts, they may prove as "wicked" a problem in small and intermediate as in larger cities.

**Author Contributions:** D.S., Y.V., T.S. and W.S. contributed to all stages of the conceptualisation, design, drafting and editing of the paper. All authors have read and agreed to the published version of the manuscript.

**Funding:** Warren Smit would like to acknowledge the support of the PEAK Urban programme, funded by the U.K. Research and Innovation Global Challenges Research Fund [ES/ P011055/1].

**Data Availability Statement:** There are no supplementary materials relating to South Africa. For India, please refer to the Nagrikal series (https://www.nagrika.org/nagrikal, accessed on 2 November 2020) by Nagrika, which analyses the constitutional devolution of functions to city governments in India and the current paradigm of multi-level urban governance in India. This article from Nagrikal refers to the classification of cities in India along with a global perspective.

**Acknowledgments:** Although the work reported here was not a part of that programme, the authors all collaborated in Mistra Urban Futures, the international research centre on urban sustainability based at Chalmers University of Technology, Gothenburg, Sweden.

**Conflicts of Interest:** The authors declare no conflict of interest.

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
