# Peer review of "Responding to Climate Change in Small and Intermediate Cities: Comparative Policy Perspectives from India and South Africa"

_sustainability, doi:10.3390/su13042382_

Round 1
Reviewer 1 Report
The manuscript examines how small and intermediate urban centres are addressing environmental challenges according to the independence degree of local authorities within the multi-level governance framework, and some Indian and South African cities are selected as case studies.
The topic is very interesting and needs further study, as small and intermediate urban centres play an important role for a sustainable development of rural regions.
Although the manuscript is well written, it looks more like a report or, at least, a review of policy and governance documents than a scientific article.
Main concerns
Introduction. Introduction lacks an adequate international review of the topic.
Method. There is not any description of a scientific method but just a reference to a generic review (lines 74 and 77) of the division of power within the different levels of the governance framework and the policy documentation dealing with environmental issues.
Results. A result of the review is Table 1 (Indian case study) that provides a summary of the jurisdiction of some small-sized Indian cities over key climate change-related functions. Unfortunately, a similar table of the other case study has not even been provided.
Discussion. Authors made several valuable comments in the Discussion and Conclusion sections, even if they are too much generic.
In my opinion original contributions of a scientific article should be more than a limited review of policy documents. The manuscript falls short of meeting the standards of the Sustainability Journal and is not publishable as it stands.
In conclusion, the manuscript has some merit and could be a very good starting point to deepen the topic and continue with the research by adding the collection of other quantitative data.
For example, in relation to the data in section 3.1.1: there are 7935 Indian census towns, but how many and which natural resources did they consume? what are the environmental impacts? In what terms could local management change the impact on environment and climate? How can it be measured? What are the different international management models that can be referred to? Which are the feasible actions? These are just some questions. It is difficult to give other suggestions because the research could be developed in many different ways.
Whatever the development, a scientific method for collecting and processing the data must be clearly described and applied to both case studies to obtain comparable results.
Reviewer 2 Report
The paper presents a very interesting issue of the place of small and intermediate cities in tackling climate change. Climate related issues are currently in the frontline of the problems challenging humankind. To handle it institutions at all the levels of public administration must act. It is vital to know what tools and powers each of the stakeholders has at the moment and what barriers for active involvement in fighting climate change it faces.
The paper shows a rarely discussed issue of the role of cities other than the biggest metropolises in tackling the problem of climate change. It shows that such cities do not have sufficient powers to effectively support their communities in tackling climate change.
The research findings clearly show significant shortcomings in the potential of small and intermediate cities in climate change adaptation. These finding should serve as recommendations for policymakers not only in India and South Africa. Cities must be empowered to play an active role in supporting other policies and policy instruments introduced to handle the problem of climate change. The paper is an important remainder of the necessity to act together and engage all the stakeholders.
The paper is clearly written and easy to read. The conclusions are consistent with the evidence presented. Thus, the findings of the study are a strong argument for increasing the involvement of the administrative levels not fully engaged in tackling climate change.
In my opinion, the paper acts as an important and much needed voice in the debate on duties and roles of different stakeholders in struggling with climate change issues.
Reviewer 3 Report
1.It is necessary to justify better why these two countries have been chosen for the comparison
2. It is indicated that the analysis will improve resilience in the abstract but this is not analyzed in the text
Reviewer 4 Report
The paper in present form is not more than a very simplified report. For example, There is not detailed methodology, scientific analysis, data used. Therefore, authors are advised to provide a detail description of used datasets in the context of climate change. Also, provide scientific analysis.
Round 2
Reviewer 1 Report
The authors have improved the manuscript and now the structure of their research is clear.
However, paragraph 3 still requires some changes to make the case study comparison more effective.
When two different case studies are presented, the approach (and also the form) must be unique in order to make evident similarities and differences. For this reason, as a scholar and a reader of the article I expect paragraphs 3.1.1 and 3.2.1 to follow the same structure and provide the same type of data, expressly declaring any absences or discrepancies. Eg. if in paragraph 3.1.1 authors write "India has a population of .... and a number of cities of… ” (on a national scale), I expect that also in paragraph 3.2.1 they write first about South Africa and not about Western Cape (on a provincial scale). The scale is different, and no comparison can be made.
I make the same remark for table 1. Authors should construct a table structure that can apply to both case studies and that contains the relevant information on jurisdictions. Even any missing data, as a result of verified absence of similar jurisdictions, are themselves a result of the research.
I appreciate the data added in table 2, e.g. Gini Index, but a similar table should also be made for the Indian cities. The goal is always the same: to make a direct comparison between the two case studies.
In conclusion, I suggest that the authors carefully modify the structure of the two paragraphs so that they are straightforwardly comparable. This will certainly improve the communication of research results.
Finally, given that Sustainability is an international journal, I suggeste to include two figures (India and South Africa) with the location of the cities of the case studies.
Author Response
Thanks for your further suggestions, which have all been implemented: where available, Gini co-efficients for the Indian cities have been added to Table 1 to match the format of Table 3 on the South African cities.
Maps have been added showing the locations of the respective cities (Fig 1 on India and Fig 2 on South Africa).
Additional text has been added to the South African introduction at the start of section 3.2.1 (lines 665-670) to provide national data.
Reviewer 4 Report
Revised version has improved.
Author Response
Thanks for acknowledging the improvements. A few additional small enhancements have been made in response to points raised by another reviewer.